# Enhanced stability and polyadenylation of select mRNAs support rapid thermogenesis in the brown fat of a hibernator

Katharine R Grabek[1,2†], Cecilia Diniz Behn[3], Gregory S Barsh[4], Jay R Hesselberth[1], Sandra L Martin[1,2]*

[1]Department of Cell and Developmental Biology, University of Colorado School of Medicine, Aurora, United States; [2]Human Medical Genetics and Genomics Program, University of Colorado School of Medicine, Aurora, United States; [3]Department of Applied Math and Statistics, Colorado School of Mines, Golden, United States; [4]Department of Research, HudsonAlpha Institute for Biotechnology, Huntsville, United States

**Abstract** During hibernation, animals cycle between torpor and arousal. These cycles involve dramatic but poorly understood mechanisms of dynamic physiological regulation at the level of gene expression. Each cycle, Brown Adipose Tissue (BAT) drives periodic arousal from torpor by generating essential heat. We applied digital transcriptome analysis to precisely timed samples to identify molecular pathways that underlie the intense activity cycles of hibernator BAT. A cohort of transcripts increased during torpor, paradoxical because transcription effectively ceases at these low temperatures. We show that this increase occurs not by elevated transcription but rather by enhanced stabilization associated with maintenance and/or extension of long poly(A) tails. Mathematical modeling further supports a temperature-sensitive mechanism to protect a subset of transcripts from ongoing bulk degradation instead of increased transcription. This subset was enriched in a C-rich motif and genes required for BAT activation, suggesting a model and mechanism to prioritize translation of key proteins for thermogenesis.

*For correspondence: sandy.martin@ucdenver.edu

**Present address:** †Department of Genetics, Stanford University School of Medicine, Stanford, United States

**Competing interests:** The authors declare that no competing interests exist.

## Introduction

Many mammals hibernate to conserve energy during extended periods of limited resource availability and harsh environmental conditions. As winter approaches in temperate climates, hibernators enter into a state of torpor. Torpor in ground squirrels involves active suppression of physiological processes to 2–5% of basal rates, which allows body temperature to lower to just above ambient, even as ambient temperatures fall to near freezing. This depressed state is not continuous throughout winter, however, instead it lasts for 1–3 weeks until it is punctuated by a spontaneous, rapid re-warming to 37°C; physiological rates during re-warming match or even exceed basal rates. The interbout arousal period is then sustained for 12–24 hr before torpor resumes. Cycles between torpor and arousal result in winter heterothermy or hibernation (*Figure 1*). Hibernation persists for 5–8 months before emergence in spring and maintenance of more typical mammalian homeostatic physiology throughout the summer period of growth and reproduction (*Figure 1*, see *Carey et al., 2003*; for review). Although of broad medical interest for their ability to tolerate these extraordinary physiological extremes (*Carey et al., 2003*; *Andrews, 2007*; *Carey et al., 2012*; *Dave et al., 2012*), many aspects of the hibernation phenotype remain poorly understood. Some of hibernation's most

**eLife digest** Many mammals hibernate to avoid food scarcity and harsh conditions during winter. Hibernation involves entering a state called torpor, which drastically reduces the amount of energy used by the body. During torpor, body temperature also decreases. This is particularly exemplified in ground squirrels, whose body temperature can hover at just above or even below the point of freezing. However, hibernating mammals cannot remain in this state continuously over the months of hibernation but instead cycle between bouts of torpor lasting for 1–3 weeks and brief periods of 'arousal' lasting between 12–24 hr, during which their body rapidly warms up.

The heat required to start warming up the hibernator is generated from a specialized form of fat called brown adipose tissue. Normally, the bursts of metabolic activity that are required to create this heat depend on certain proteins being produced. Making a protein involves 'translating' its sequence from template molecules called messenger RNA (mRNA), which are 'transcribed' from the gene that encodes the protein. During the low body temperatures experienced during torpor, both of these processes stop. So how is the hibernator able to quickly and efficiently heat itself up during the arousal periods of hibernation?

Grabek et al. investigated this by analyzing the relative levels of mRNA in the brown adipose tissue of hibernating 13-lined ground squirrels. Using a special technique to sample and sequence small fragments of mRNA taken from brown adipose tissue, Grabek et al. compiled a profile of the mRNA molecules present at different points in the torpor–arousal cycle and compared this with a similar profile taken from squirrels that were not hibernating.

From this analysis, Grabek et al. detected that a particular group of mRNA molecules that are required for producing heat increase in abundance during torpor, even though body temperature is low enough to stop gene transcription. This increased abundance does not occur because more of the mRNA molecules are made; instead, the mRNA molecules are modified to become more stable and long lasting. Once the animal warms up during arousal, gene transcription is reactivated and more new mRNA molecules are made.

Grabek et al. suggest that the key mRNAs required for brown adipose tissue function are selectively stabilized during torpor through a temperature-dependent protective mechanism. These mRNAs are then preferentially translated into proteins during arousal to rapidly and efficiently heat the hibernator. Most other mRNA molecules degrade throughout torpor, and so their numbers decline as replacements are not transcribed until body temperature briefly recovers during arousal. Whether this protective mechanism is also used in other tissues during torpor remains a question for future work.

defining mysteries are the mechanisms that underlie the highly dynamic oscillations of the torpor–arousal cycle.

Transcription and translation effectively cease at low body temperature during hibernation (*van Breukelen and Martin, 2001*; *van Breukelen and Martin, 2002*), yet organs maintain integrity and in some cases are quickly reactivated after 2 weeks of near inactivity in torpor. The need for immediate intense metabolic activation at low temperature is most pronounced in brown adipose tissue (BAT); early re-warming depends exclusively on non-shivering thermogenesis in this organ (*Cannon and Nedergaard, 2004*). Because of the constraints on gene expression during torpor, the rapid burst of metabolic activity that characterizes early re-warming may be particularly challenging for BAT.

To balance the decreased transcription, mRNA degradation during torpor also must be reduced to maintain cellular integrity and permit function in early arousal. Just as with transcription, low body temperature (i.e., Q10 effects) will slow rates of RNA degradation (*Burka, 1969*; *Bremer and Moyes, 2014*), but it is unclear how these two opposing activities will converge after two weeks to determine the steady-state abundance of specific RNAs at the end of a torpor bout. While the general consensus is that the transcriptome is largely stable during torpor (reviewed by *Tessier and Storey (2014)*), this view is based upon results (*Frerichs et al., 1998*; *O'Hara et al., 1999*; *Knight et al., 2000*; *Williams et al., 2005*) where ongoing degradation is not readily distinguishable from a stable transcriptome because of the sampling and normalization strategies employed. There are few clear examples of transcripts that diminish across a torpor bout (*Epperson and Martin, 2002*) and others that appear to

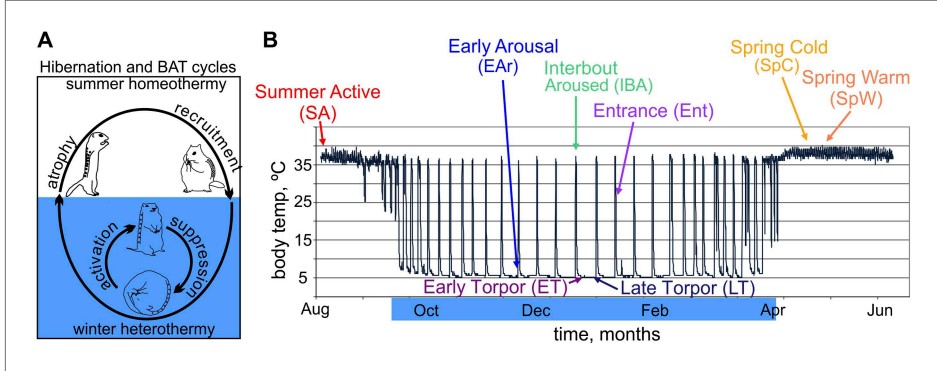

**Figure 1**. The hibernating phenotype as a model for studying BAT metabolic regulation. (**A**) Schematic depicting the metabolic suppression and activation cycle of BAT during the highly recruited, winter hibernation phase (blue shading) of the annual cycle. Cartoon squirrels represent general phenotypic changes among annual and torpor–arousal cycles (*Hindle and Martin, 2014*). (**B**) Relationship of sample groups to body temperature over time. Blue highlighting on months indicates hibernation.

increase (*O'Hara et al., 1999*), largely because few studies provide the necessary temporal resolution to quantify changes across a torpor bout.

In this study, we interrogate BAT mRNA dynamics in 13-lined ground squirrels across the torpor–arousal cycle and the circannual rhythm of hibernation (*Figure 1B*). We chose BAT because of its unique requirement to function quickly and maximally in the earliest moments of arousal, after spending two weeks at the transcriptionally prohibitive body temperatures of torpor. We used a transcriptional pro-filing approach developed for non-model organisms, EDGE (*Hong et al., 2011*), on five precisely timed sample groups to capture multiple phases of the torpor–arousal cycle (*Figure 1B*) and, for com-parison, three groups from the non-hibernating, homeothermic portion of the year (*Figure 1B*).

## Results

A total of 38 EDGE-tag libraries, representing 8 distinct sampling groups (*Figure 1B*), were sequenced, processed (*Figure 2—figure supplement 1*), and analyzed for changes associated with hibernation physiology. For each of the libraries, 90.1 ± 2.6% of the sequence reads aligned to ground squirrel genomic (Supplementary file 3A in *Grabek et al., 2014*) or mitochondrial DNA (*Figure 2A*). After normalization, filtering, and annotation (*Figure 2B*, *Figure 2—figure supplement 1*), 14,798 EDGE-tags representing 8,089 unique genes remained (Supplementary file 3B in *Grabek et al., 2014*). We first clustered the individual sample libraries by tag abundance using Random Forests (*Breiman, 2001*). Three main groups were evident (*Figure 2C*): (1)'spring', independent of ambient temperature, spring cold (SpC), and spring warm (SpW); (2) 'winter warm': interbout aroused (IBA), entrance (Ent), and summer active (SA); (3) 'winter cold': early torpor (ET), late torpor (LT), and early arousal (EAr). Notably, BAT samples harvested from winter animals at warm body temperature clus-tered separately from those at low body temperature. This separation indicates the transcriptome is dynamic across a torpor bout.

The 14,798 tags were next tested for significant differential expression among the three main groups; changes were detected in 2,159 tags (14.6%; q < 0.05) representing 1,638 unique genes (Supplementary file 3C in *Grabek et al., 2014*). These correlated well with quantitative changes in the BAT transcriptome of this species reported previously (*Hampton et al., 2013*); 91% of overlapping differentially expressed transcripts exhibited changes in the same direction among comparable states (see 'Materials and methods'). DIANA hierarchical clustering identified six expression patterns among the differentially expressed tags (*Figure 2D* and Supplementary file 3C in *Grabek et al., 2014*); those in Clusters 1 and 2 were generally increased in spring compared to winter, while those in Clusters 3–6 were increased in winter, particularly in late torpor and early arousal. Distinct from the spring-enriched tags, those increased in winter were overwhelmingly enriched (*Huang da et al., 2009*) for functions related to BAT activation, such as lipid metabolism, lipid droplet formation, lipid transport, mitochon-dria and the TCA cycle (*Table 1* and Supplementary file 3D in *Grabek et al., 2014*).

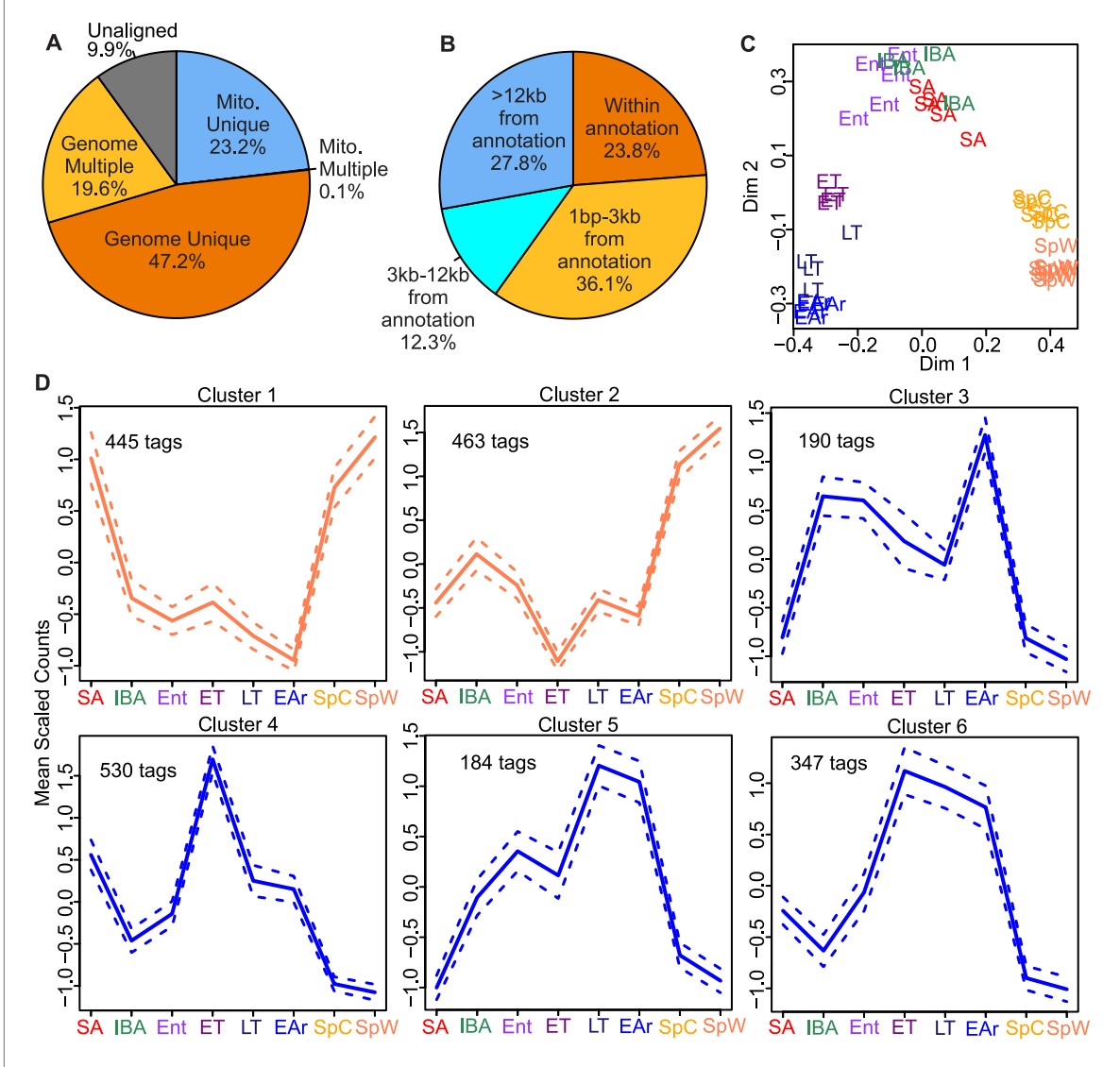

**Figure 2**. EDGE-tag library properties. Pie charts of EDGE tags mapped: (**A**) uniquely to 13-lined ground squirrel nuclear (Genome Unique) or mitochon-drial (Mito. Unique) DNA; multiple locations (Genome or Mito. Multiple); or unmapped (Unaligned); or (**B**) the indicated distances from the nearest annotated Ensembl feature (either overlapping or 3′ to the feature in kilobases). (**C**) Two-dimensional scaling plot showing Random Forests (RF) clustering of individual samples labeled by group symbol: spring warm (SpW), spring cold (SpC), summer active (SA), interbout aroused (IBA), entrance (Ent), early torpor (ET), late torpor (LT), and early arousal (EAr), as depicted in *Figure 1*. (**D**) Line plots of EDGE-tag expression patterns for all 2,159 significant differentially expressed tags; mean scaled counts, solid line, ±SEM, dotted line. Total tags in each cluster are indicated.

The following figure supplement is available for figure 2:

**Figure supplement 1**. Schematic illustrates library sequencing, read processing, tag annotation, and filtering after the creation of the EDGE-tag transcriptome libraries (see 'Materials and methods').

Surprisingly, a preponderance of winter-increased tags (i.e., transcripts) reached their highest rela-tive abundance during early torpor, late torpor, and/or early arousal (*Figure 2D*, Clusters 4–6) despite near cessation of transcription in hibernators at low body temperature (*van Breukelen and Martin, 2002*). One clue to resolve this apparent paradox was provided by *RPPH1*, the RNA subunit of RNaseP, whose relative abundance increased several 100-fold by early arousal (q < 10^−14, *Figure 3A*). Because *RPPH1* is transcribed by Pol III, it is not typically polyadenylated (*Baer et al., 1990*) and should not have been recovered in these sequencing libraries. Nevertheless, *RPPH1* acquired a long

**Table 1.** DAVID functional annotations for each DIANA cluster

| DIANA cluster | Functional annotation cluster | Enrichment score | Annotations, *n* | Genes, *n* |
|---|---|---|---|---|
| 1 | Cytosolic ribosome | 4.09 | 11 | 12 |
| | Zinc finger, C2H2-type | 2.19 | 8 | 30 |
| | Heme | 1.8 | 3 | 8 |
| | Ribosome biogenesis | 1.68 | 4 | 9 |
| | Transcription | 1.67 | 4 | 57 |
| 2 | Transcription | 6.03 | 4 | 80 |
| | Nuclear lumen | 5.91 | 4 | 57 |
| | RNA recognition motif, RNP-1 | 5.18 | 3 | 18 |
| | mRNA processing | 3.39 | 4 | 17 |
| | Transcription from RNA polymerase II promoter | 3.21 | 3 | 17 |
| 3 | Mitochondrion outer membrane | 3.07 | 4 | 6 |
| | Triglyceride biosynthetic process | 1.78 | 12 | 3 |
| | Glutathione S-transferase, C-terminal-like | 1.41 | 4 | 3 |
| | Long-chain fatty acid transport | 1.37 | 5 | 3 |
| 4 | Mitochondrial membrane | 5.84 | 30 | 30 |
| | Endoplasmic reticulum membrane | 3.94 | 20 | 20 |
| | Lipid particle | 2.73 | 5 | 5 |
| | Glucose metabolic process | 2.16 | 11 | 11 |
| | Peroxisome | 2.15 | 9 | 9 |
| 5 | Mitochondrion | 5.61 | 24 | 24 |
| | Generation of precursor metabolites and energy | 2.66 | 14 | 14 |
| | Lipid droplet | 2.38 | 3 | 3 |
| | Lipid metabolism | 2.31 | 7 | 7 |
| | Oxidative phosphorylation | 2.25 | 4 | 4 |
| 6 | Mitochondrion | 4.04 | 17 | 27 |
| | Neutral lipid biosynthetic process | 2.74 | 14 | 4 |
| | Glucose metabolic process | 2.54 | 3 | 10 |
| | Lipid catabolic process | 2.33 | 13 | 11 |
| | Adipocytokine signaling pathway | 2.17 | 11 | 9 |

The top five Functional Annotation Clusters, ordered by enrichment score (>1.3), are listed for each DIANA cluster. See *Figure 2D* for DIANA clusters and Supplementary file 3D in *Grabek et al., 2014* for all Functional Annotation Clusters.

poly(A) tail at low body temperature (*Figure 3B,C*), explaining its presence in the libraries and increase in the cold.

We considered three potential mechanisms that might explain increased transcript abundance at low body temperature: (1) elevated transcription; (2) relative stabilization; and (3) acquisition of a poly(A) tail. To probe these mechanisms, we quantified abundance and the effect of poly(A) tail length on the dynamics of *RPPH1* and thirteen other transcripts, including three additional ncRNAs and ten mRNAs (*Supplementary file 1A*; note that there are two isoforms of *LIPE*), during the torpor–arousal cycle. The absolute abundance of these transcripts was measured by RT-qPCR in total RNA, and short and long poly(A) RNA fractions (*Figure 4—figure supplement 1*; *Supplementary file 1A*) from inter-bout aroused, late torpor, early arousal, and spring warm animals (*n* = 3). Two classes of RNA dynamics were apparent; transcripts were either decreased (labeled as Class I) or stabilized (labeled as Class II) during torpor but not newly transcribed.

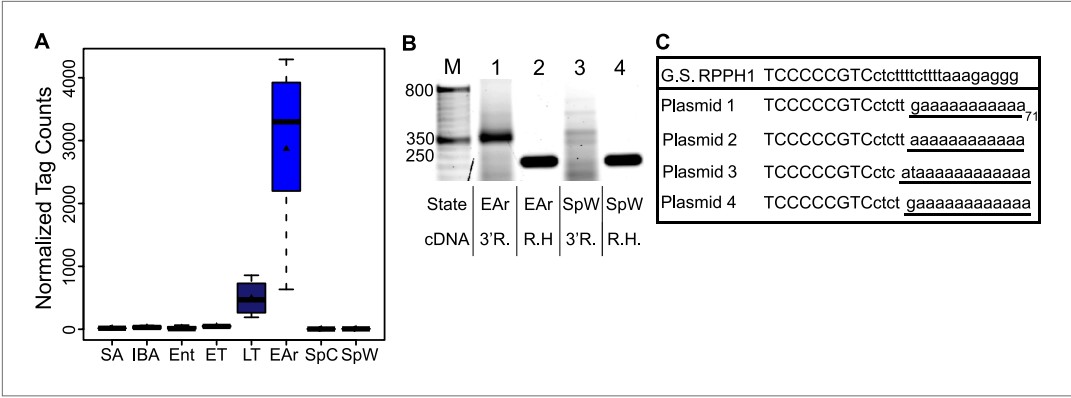

**Figure 3**. Increased RPPH1 abundance is explained by the addition of a poly(A) tail. (**A**) Box plot of normalized tag counts for *RPPH1* by state, triangle marks the mean. (**B**) Gel showing *RPPH1* RT-PCR products from 3′ RACE (3′ R, lanes 1 and 3) and random hexamer (RH, lanes 2 and 4) primed cDNA from early arousal (EAr; lanes 1–2) and spring warm (SpW; lanes 3–4) total RNA. Marker sizes are indicated on the left (50-bp ladder, lane M). (**C**) Multiple alignment of the *RPPH1* genomic DNA and four 3′ end cDNA sequences from cloned 3′ RACE (EAr) products in B: uppercase, annotated *RPPH1* RNA; lowercase, genomic DNA; underline, non-templated nucleotides.

Five Class I transcripts decreased during torpor, with poly(A) and total RNA mirroring the abundance of their EDGE tags (compare IBA to LT in *Figure 4A*; see also *Figure 4—figure supplement 2A* and *Supplementary file 1B*). Interestingly, during early arousal, when core body temperature was still low, some of these transcripts increased, likely because heat generated early in the arousal process has returned BAT to a temperature permissive for transcription (*Osborne and Hashimoto, 2003*). These transcripts were largely bearing long poly(A) tails, which also appeared to shorten during torpor (*Figure 4—figure supplement 2B*). Class I dynamics explain the DIANA Clusters 1–3, where RNA decreased during torpor but then increased at the elevated body temperature of interbout arousal (compare IBA to LT in *Figure 2D*), and likely the even larger collection of transcripts that were not differentially expressed (e.g., *GAPDH*, *Figure 4—figure supplement 2A*). Thus, it appears that most BAT transcripts slowly degrade over two weeks in torpor and are not replenished until body temperature recovers during the short euthermic period.

Nine Class II transcripts were stabilized during torpor. While their EDGE-tags appeared to increase during torpor (*Figure 4B,C*, left), this increase was not mirrored in total RNA (*Figure 4B,C*, middle-left). We further sub-divided this class by differences in polyadenylation. Total RNA for five Class IIA transcripts remained stable among states (*Figure 4B*, middle-left; *Figure 4—figure supplement 3*). However, these transcripts increased in the short and long poly(A) fractions during late torpor and particularly early arousal (*Figure 4B*, right; *Figure 4—figure supplement 3*, *Supplementary file 1C*) with concurrent poly(A) tail lengthening (*Figure 4—figure supplement 2C*), correlating with their EDGE-tags (*Figure 4B*, left; *Figure 4—figure supplement 3*; *Supplementary file 1B*). Total RNA decreased in torpor and early arousal for four transcripts in Class IIB (*Figure 4C*, *Figure 4—figure supplement 4*), whereas their polyadenylated fraction remained stable (*Figure 4C*, middle-right; *Figure 4—figure supplement 4*), resulting in an apparent increase (*Figure 4C*, right; *Figure 4—figure supplement 2D and 4*; *Supplementary file 1C*) and consistent with the EDGE-tag pattern (*Figure 4C*, left; *Figure 4—figure supplement 4*; *Supplementary file 1B*). Thus, in contrast to Class I, Class II transcripts are stabilized throughout torpor with maintenance or acquisition of a poly(A) tail. The enhanced stability of this subset relative to all other transcripts apparently leads to their relative increases at low body temperature (*Figure 2D*, DIANA Clusters 4–6).

We next tested whether the observed transcript dynamics in torpor–arousal cycles could impact the corresponding protein by measuring PNPLA2. Three PNPLA2 protein isoforms, whose sizes were consistent with those predicted for mouse in UniProt (*UniProt Consortium, 2014*), were detected by Western blot (*Figure 4D*). All appeared to cycle, but only the 48-kD band changed significantly (*Figure 4E,F*), following the dynamics of the transcript with the long poly(A) tail despite no change in overall transcript abundance (*Figure 4F*). Hence, the dynamics of this PNPLA2 protein isoform appears to be explained by polyadenylation changes in its transcript.

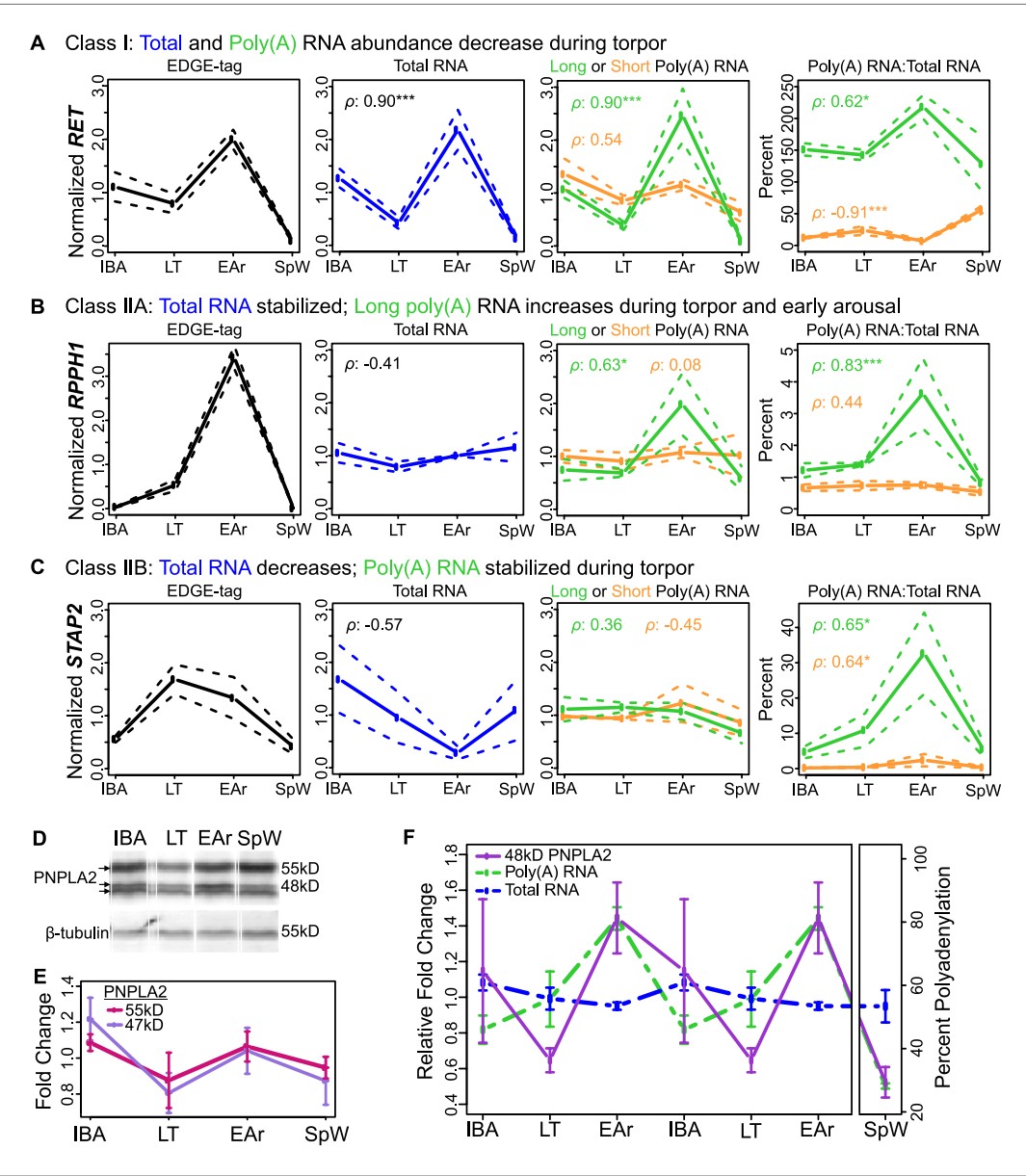

**Figure 4**. Bulk RNA degradation with stabilization of selected transcripts and cycles of re-adenylation at low body temperature. (**A**) Class I, represented by *RET* proto-oncogene. Relative expression levels (solid line; ±SEM, dotted line; y-axis) of EDGE-tag counts (far left), total RNA (middle-left), poly(A) RNA (green = long poly(A) RNA; orange = short poly(A) RNA; middle-right), and percent recovery (y-axis) of poly(A) RNA relative to total RNA (far right) among physiological states: interbout aroused (IBA), late torpor (LT), early arousal (EAr), and spring warm (SpW). Spearman correlations (ρ) to EDGE-tag expression labeled in three right boxes; *p ≤ 0.05, **p ≤ 0.01, ***p ≤ 0.005. (**B–C**) Labeling is as in panel **A**. (**B**) Class IIA, represented by *RPPH1*. (**C**) Class IIB, represented by *STAP2*. (**D**) Western blot reveals three isoforms of the PNPLA2 protein (left arrows) among indicated (top) sample states; marker sizes are denoted on right, β-tubulin, below, served as a loading control. (**E**) Relative abundance (solid lines; ±SEM, bars) of the 55 and 47 kD PNPLA2 proteins among samples states. (**F**) Relative abundance pattern of the PNPLA2 48 kD protein, *PNPLA2* long poly(A) and total RNA; hibernation states are double-plotted to reveal cyclical pattern of torpor and arousal.

The following figure supplements are available for figure 4:

**Figure supplement 1**. ePAT confirmation of RNA fractionation by poly(A) tail length.

*Figure 4. Continued on next page*

*Figure 4. Continued*

**Figure supplement 2**. Class I RNA and poly(A) tail dynamics.

**Figure supplement 3**. Class IIA RNA dynamics.

**Figure supplement 4**. Class IIB RNA dynamics.

---

To investigate how changes in rates of transcription and degradation could affect differential gene expression in torpor, we developed a mathematical model of transcript dynamics across the torpor–arousal cycle. We simulated a population of 50 'protected' transcripts and a bulk population of 1,400 transcripts; these numbers are proportional to the 531 tags that were either increased or stabilized across a bout of torpor (DIANA Clusters 5 and 6, *Figure 2D*) compared to the 14,267 tags in the full dataset. For this simulated population, the abundance of each RNA transcript was governed by a differential equation describing temperature-dependent rates of RNA synthesis and degradation (*Schwanhausser et al., 2011*). To model RNA transcript dynamics across the torpor–arousal cycle (*Figure 5—figure supplement 1*), we introduced a representative 12-day body temperature profile, incorporating temperature-dependence into the rates of RNA synthesis and degradation based on Q10 effects (*Burka, 1969*; *van Breukelen and Martin, 2002*), as described in detail in *Supplementary file 2*.

In the 50-transcript subset, we implemented either fixed or temperature-dependent alterations to degradation and synthesis rates to determine the resulting protective effects on normalized transcript abundance following 10 days of torpor. We found that a temperature-dependent mechanism that protected a subset of transcripts relative to bulk RNA degradation (*Figure 5A,B*) was most consistent with the increased abundances observed experimentally. For a body temperature threshold of 10°C and degradation set to 3% of its rate in the warm animal, the relative abundance of the protected transcripts increased over twofold (*Figure 5C,D*), best reflecting the experimental data. This effect was dose-dependent with the level of protection and was relatively insensitive to thresholds above 10°C (*Figure 5E*). Although temperature-independent decreases in degradation rates also led to increases in the relative abundance of protected transcripts, this mechanism required implausible compensatory changes to either steady state RNA abundance or transcription rates in the warm animal (*Figure 5—figure supplement 2*). Due to the differential Q10 effects on transcription and degradation, increasing transcription rate did not produce relative abundance increases (*Figure 5—figure supplement 3*). Thus, in agreement with RT-qPCR data, mathematical modeling supports enhanced stabilization of a subset of transcripts via a temperature-dependent protective mechanism; this, rather than increased transcription, leads to the observed increase in their relative abundances at the low body temperature of torpor.

Finally, to address the possible mechanism underlying protection of selected transcripts from degradation, we examined transcript 3′ untranslated region (UTR) sequences for shared motifs (*Figure 6A*); because ground squirrel 3′ UTRs are largely unannotated, we defined 3′ UTRs as the 500 nt region immediately downstream of the stop codon. This choice was validated by taking a random sampling of 3′ UTR sequences in all clusters, which returned enrichment for a motif resembling the polyadenylation signal (*Colgan and Manley, 1997*) when compared to a scrambled background set (*Figure 6B*; [*Bailey et al., 2009*]). To identify motifs unique to the protected RNA subset, the significantly changed transcripts were divided into two groups for comparison (*Figure 6A*): (1) the positive set of transcripts that appeared to be stabilized (e.g., *Figure 2D*, DIANA Cluster 6) or increased in torpor (e.g., *Figure 2D*, DIANA Clusters 4–5); and (2) the negative set of transcripts that appeared to decrease in torpor (e.g., *Figure 2D*, DIANA Clusters 1–3). When the positive set was compared to the negative set, EXTREME (*Quang and Xie, 2014*) identified two significantly enriched C-rich motifs (Motif 1 and 2, *Figure 6C,D*). Significantly, DIANA Cluster 5, comprised of the transcripts that most clearly increased in relative abundance from early to late in torpor, contained the greatest percentage of mRNAs with the two motifs (Motif 1: 59.1%; Motif 2: 66.7% of transcripts, *Figure 6C,D*). DIANA Cluster 6, comprised of transcripts that remained elevated and stable across a torpor bout, contained a higher proportion of transcripts with Motif 2 (50.8%, *Figure 6D*) relative to the other DIANA clusters (8.5–39.8%, *Figure 6D*) and the control, non-significant transcripts (21.1%, NS in

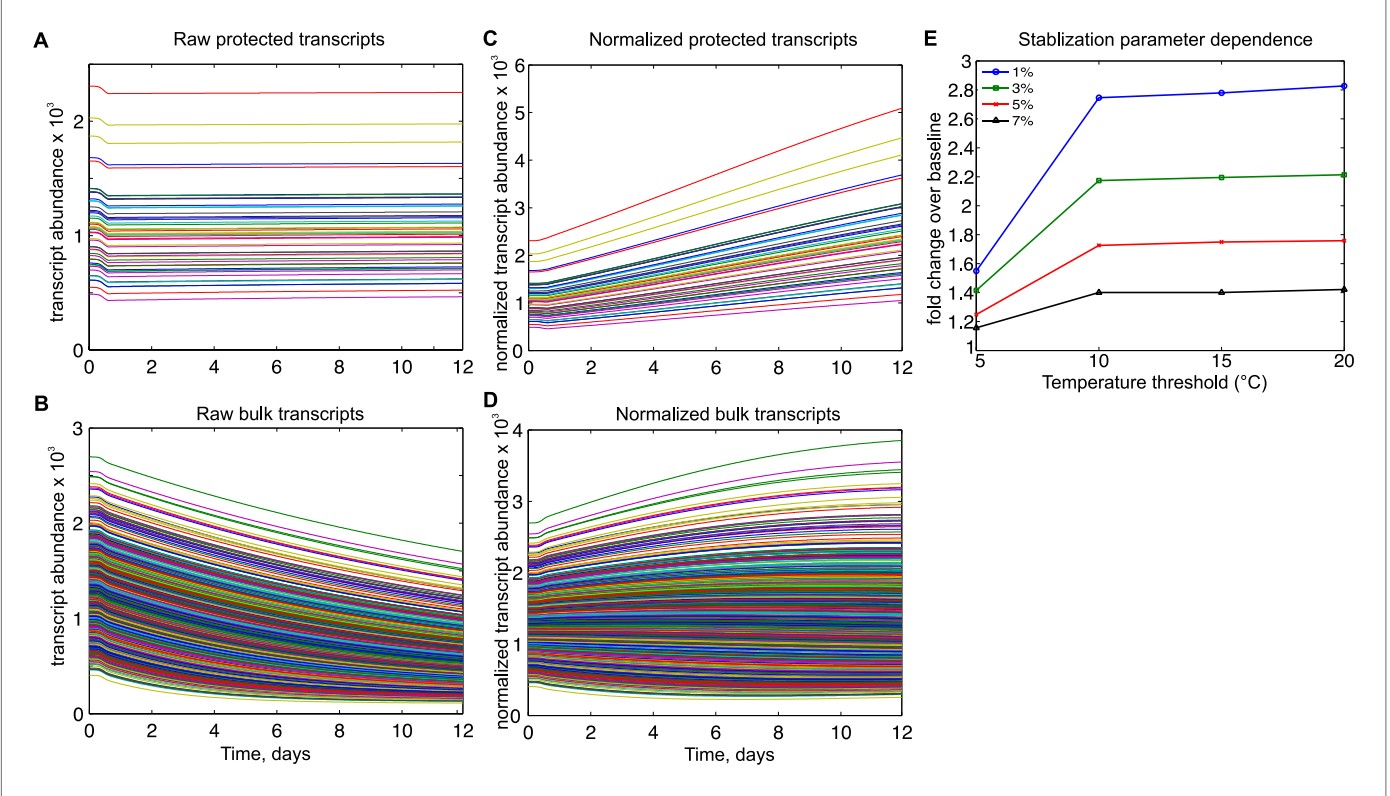

**Figure 5**. Mathematical modeling dynamics for 1,400 bulk and 50 protected transcripts simulated over the 12-day torpor–arousal cycle. A temperature-dependent protective mechanism against degradation is implemented for protected transcripts: for body temperature below 10°C, degradation is set to 3% of its rate in the warm animal. Transcription rates for both population and degradation rates for bulk transcripts are adjusted for Q10 effects. Low body temperature during torpor causes the raw abundance of both (**A**) protected and (**B**) bulk transcripts to decrease. When abundances are normalized across the population, (**C**) protected transcripts appear to increase approximately twofold while the majority of (**D**) bulk transcripts still appear to degrade. (**E**) Systematically varying the temperature threshold and the percentage of the warm degradation rate associated with the protective mechanism reveals a dose-dependent relationship in which higher temperature thresholds and lower percentages are associated with larger fold increases over baseline in the protected subset.

The following figure supplements are available for figure 5:

**Figure supplement 1**. Mathematical modeling of transcript degradation.

**Figure supplement 2**. Results for Mechanism 2.

**Figure supplement 3**. Results for mechanism 1.

*Figure 6D*). Additionally, the other winter-increased DIANA Clusters, 3 and 4, were relatively enriched for these motifs as compared to the spring-increased DIANA Clusters 1 and 2 (38.4–43% vs 8.5–23.3%, *Figure 6C,D*), suggesting that these motifs play a broader role in enhanced transcript stability and/or translation during winter heterothermy. Although the transcripts in DIANA Cluster 4 appeared elevated in early torpor (*Figure 2D*), their motif enrichment was similar to that of Cluster 3. In contrast to those in Clusters 5 and 6, these transcripts also appeared to largely degrade by late torpor (compare ET to LT, *Figure 2D*); hence their reduced motif enrichment is consistent with reduced transcript stability across a bout of torpor.

To identify putative binding protein(s) for these enriched sequence motifs, we used TOMTOM (*Gupta et al., 2007*), searching against a database of RNA binding motifs (*Ray et al., 2013*; *Ji et al., 2013*). Our motifs significantly matched those reported by *Ji et al. (2013)* (*Figure 6E*), implying binding by a poly(C) binding protein. We detected expression of two poly(C) binding protein paralogs in our dataset: *PCBP3* and *PCBP4*. While *PCBP4* did not vary with hibernation physiology, *PCBP3*

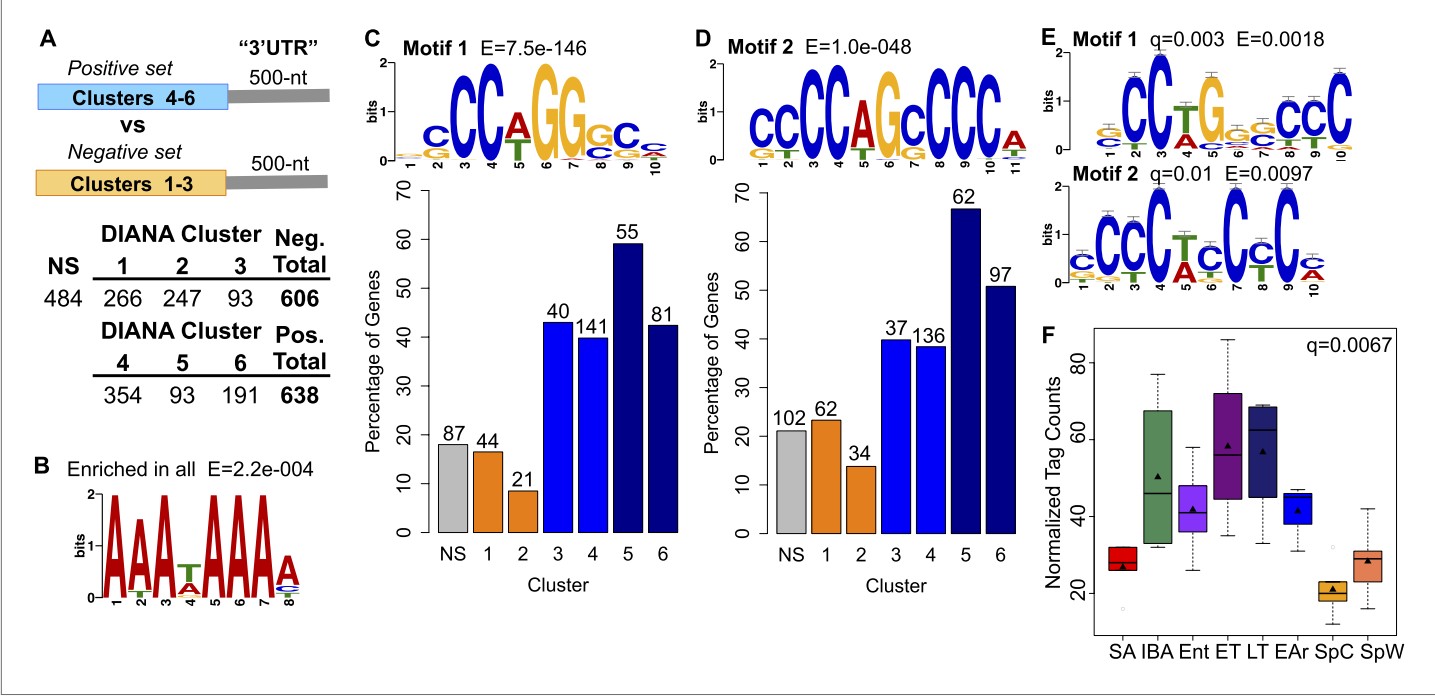

**Figure 6**. Motif enrichment in the 3′ ends of protected transcripts. (**A**) Schematic shows methodology for identifying motifs enriched in the 3′ UTR regions (500-nt) of transcripts increased or stabilized in torpor (positive set; transcripts in DIANA Clusters 4–6) compared to transcripts decreased in torpor (negative set; transcripts in DIANA Clusters 1–3). The table below lists the number of unique transcripts within each DIANA Cluster used in the analysis, the sum of those in the negative or positive set and the number of non-significantly changed transcripts (NS) used in later comparisons. (**B**) The motif closely resembling the AAUAAA polyadenylation signal (*Colgan and Manley, 1997*) identified by a random sampling of 3′ UTR sequences in all clusters compared to a scrambled background set. (**C**) Motif 1: the most significant motif identified in the positive set of transcripts when compared to the negative set. Bar plot below shows the percentage of transcripts in each DIANA Cluster or in the non-significant group (NS) that contains Motif 1. Actual numbers of transcripts containing Motif 1 are labeled above bars (see table in **A** for comparison). (**D**) Motif 2: the second and the only other significant motif identified in the positive set of transcripts when compared to the negative set. Labeling is the same as in **C**. (**E**) The top RNA-binding protein motif matches for Motif 1 and 2 using TOMTOM (*Gupta et al., 2007*). These correspond to the poly(C) binding protein motifs reported by *Ji et al. (2013)*. (**F**) Box plot of normalized tag counts for *PCBP3* by state, triangle marks the mean.

expression increased significantly in the winter groups (q = 0.0067, *Figure 6F*). The enrichment of PCBP binding motifs in the subset of transcripts that increased at low body temperature, together with the increased *PCBP3* abundance in winter, suggest a role for PCBP3 in protecting a subset of BAT transcripts via its binding to the 3′ UTR C-rich motifs during torpor.

## Discussion

Our results show that enhanced stabilization and polyadenylation of a crucial group of transcripts, with evidence of ongoing bulk RNA degradation, occur during torpor and are likely tied to rapid activation of BAT. Dynamic polyadenylation has been demonstrated to control temporal and spatial regulation of translation in many systems (*Wu et al., 1998*; *Kojima et al., 2012*), including maternal *Xenopus* oocyte maturation (reviewed in *Vasudevan et al. [2006]*). Generally, poly(A) tail elongation causes translational activation, whereas shortening leads to silencing and/or RNA degradation (*Weill et al., 2012*). Similar to oocyte maturation, in hibernation, the transcriptional machinery is silenced during the two week period of torpor; therefore, post-transcriptional mechanisms affecting mRNA stability and translation serve in the rapid switch between a hypo- and hyper-metabolic state in BAT. For instance, PNPLA2 catalyzes the first committed step in triacylglycerol hydrolysis, resulting in diacyl-glycerol and free fatty acid (*Zimmermann et al., 2004*). Its increased translation via poly(A) tail length-ening in early arousal would ensure immediate generation of free fatty acids for thermogenesis, while poly(A) shortening during interbout arousal offers a parsimonious means to silence protein translation and lower free fatty acids as metabolic activity declines. Additionally, dynamic polyadenylation likely

controls translation of the other Class IIA transcripts. Although there is currently no evidence for this type of mechanism operating in human BAT, our results suggest a potential therapeutic strategy by which translation of key proteins can be prioritized for recovery from metabolic repression, and more specifically, for rapid activation of thermogenesis in BAT.

A role for post-transcriptional regulation in hibernation was posited previously, based on poor correlations between mRNA and protein levels (*Shao et al., 2010*). In this study, we provide evidence for stabilization of a specific, functionally relevant subset of mRNAs during torpor. Moreover, our findings are consistent with reports of mRNA degradation during torpor (*Epperson and Martin, 2002*) and with global maintenance of poly(A) tails and their increased length (*Knight et al., 2000*). These results also provide further explanation to histological observations, as RNP granules containing both ncRNAs and mRNAs are formed during torpor in BAT nucleoli (*Malatesta et al., 2008*; *Malatesta et al., 2011*). The Class II ncRNAs identified in our study are located in nucleoli, suggesting that sequestration into these RNP granules protects a subset of transcripts from the degradation affecting bulk BAT RNA, which is also consistent with the results of our mathematical modeling.

Our results propose that a mechanism underlying enhanced stabilization of a subset of RNAs involves their binding by one of the poly(C) binding proteins (PCBPs), as we detected their corresponding C-rich motifs in the 3′ regions of the torpor-stabilized mRNAs. Intriguingly, PCBPs are involved in many aspects of post-transcriptional control that are consistent with our observations, including enhanced stability of long-lived mRNAs (*Makeyev and Liebhaber, 2002*). Further roles include 3′ end-processing and alternative polyadenylation (*Ji et al., 2011*; *Ji et al., 2013*), and the addition, maintenance (*Wang et al., 1999*), and elongation of poly(A) tails (*Vishnu et al., 2011*). Finally, these proteins are involved in both translational silencing and enhancement (*Makeyev and Liebhaber, 2002*). Although these roles are established for the predominantly studied PCBP1 and PCBP2, these paralogs were not detected in our dataset. Rather, we detected increased expression of *PCBP3* during winter heterothermy, a pattern that would be expected for a role involving enhanced mRNA stabilization and polyadenylation during torpor. Furthermore, *PCBP3's* pattern runs in contrast to most of the other RNA binding proteins detected in this dataset, which, if changed, were largely decreased during winter heterothermy (see *Table 1*, functional annotations for DIANA Clusters 1 and 2).

In addition to a PCBP, other 3′ UTR binding proteins may be involved in enhancing mRNA stability during torpor. The poly(A) binding protein PABP1 and the TIA-1/R RNA binding proteins were recently shown to localize to discrete sub-nuclear foci during torpor in the livers of 13-lined ground squirrels (*Tessier et al., 2014*). PABP1 specifically binds to the poly(A) tails of transcripts, influencing their length as well as overall transcript translation and stability (*Burgess and Gray, 2010*). Significantly, the PCBPs 1 and 2 functionally interact with PABP1 in order to prevent deadenylation and to maintain mRNA stability (*Wang et al., 1999*). While further research is needed to determine whether the protection identified here extends to other tissues during torpor and to thoroughly examine the role of PCBP3 or its homologs in this protection, our results of enhanced stabilization and polyadenylation for a subset of crucial transcripts suggest a mechanistic link to PABP1 observations in other organs (*Knight et al., 2000*; *Tessier et al., 2014*). Although there are reports of elevated RBM3, a cold-induced RNA binding protein, in several organs including BAT from ground squirrels and bears during hibernation (*Williams et al., 2005*; *Yan et al., 2008*; *Fedorov et al., 2011*), we found no evidence for enrichment of RBM3 recognition motifs (*Liu et al., 2013*; *Ray et al., 2013*) in our subset of stabilized transcripts.

More broadly, our study highlights the importance of how transcriptome data is interpreted. While it is generally assumed that changes in steady-state mRNA levels stem from changes in the rate of transcription, varying the rate of degradation also changes steady-state levels; this phenomenon has been observed at both mRNA and protein levels in the cold acclimation of fish (*Sidell, 1977*; *Bremer and Moyes, 2014*). Recently, the balance between mRNA synthesis and degradation rates was examined on a global scale in cultured mammalian cells, demonstrating complex gene-specific effects of transcription, processing, decay, and translation (*Rabani et al., 2011*; *Schwanhausser et al., 2011*). In our study, the modeling results shown in *Figure 5* demonstrate that transcript-specific variability in the rate of degradation may cause a subset of transcripts to increase in relative abundance. These transcripts are degrading, but more slowly than most of the bulk transcripts and therefore exhibit a small increase in relative abundance across the torpor bout. Thus, many of the smaller fold changes observed in gene expression datasets from hibernators (*Williams et al., 2005*; *Yan et al., 2008*; *Hampton et al., 2013*; *Schwartz et al., 2013*) likely reflect intrinsic differences in the stability of specific mRNAs rather

than specific mechanisms to regulate their transcription or decay. However, for changes greater than approximately twofold in a substantial number of transcripts, here ~3.5% of the total, a specific regulatory mechanism appears to be required. Furthermore, particularly large fold changes, as observed here with *RPPH1*, likely reflect the addition or lengthening of poly(A) tails.

Our data suggest a model (*Figure 7*) of RNA dynamics in hibernator BAT wherein key RNAs for BAT function are selectively stabilized during torpor while bulk transcripts decline through degradation in the absence of new transcription. Stabilization likely occurs by a temperature-dependent protective mechanism that is in place before body temperature reaches 5°C, such as PCPB3 binding to the 3′ UTRs of protected transcripts, which then leads to their relative increase as torpor progresses. At the end of torpor and onset of arousal, the stabilized mRNA subset with the longest poly(A) tails is translated immediately as BAT temperature becomes permissive. As BAT temperature rises, further polyadenylation of the remaining stabilized RNAs facilitates their translation, transcription resumes (*Osborne et al., 2004*), and, during interbout arousal, transcripts that were previously degraded during torpor are replenished to their baseline levels. This dynamic cycle of transcription, degradation, stabilization, and polyadenylation in BAT leads to translation of the correct transcripts at the correct time with minimal energy expenditure. Specifically: (1) energy intensive translation during early arousal is directed to proteins needed for BAT activation; (2) the cell is not dependent on de novo transcription at the onset of the short bursts of metabolic activity, which could delay thermogenesis and induce stress; (3) inhibition of translation via shortening of poly(A) tails while body temperature is high or begins to decline conserves energy compared to mRNA degradation and subsequent re-synthesis. Thus, given a general suppression of transcription by low body temperature during two-week torpor periods, stabilization and dynamic polyadenylation provide an alternative mechanism to prioritize transcripts for immediate translation when BAT metabolic activity rapidly resumes.

## Materials and methods

### Animal and tissue collection

13-lined ground squirrels were procured and housed as described previously (*Hindle and Martin, 2014*). All animals except those in the summer active group (SA; *n* = 5; *Figure 1B*) were surgically implanted in late August or early September with both an intra-peritoneal datalogger (iButton, Embedded Data Systems) and a radiotelemeter (VM-FH disks, Mini Mitter, Sunriver, OR) for remote body temperature monitoring until tissue collection. All animal protocols were approved by the University of Colorado Institutional Animal Care and Use Committee.

BAT samples were collected from the axillary pads in animals representing eight different seasonal and physiological groups. All groups with their approximate body temperature and time of year are depicted in *Figure 1B*. Five groups represent animals in the winter hibernating portion of the year, including: early torpor (ET; Tb = 4°C for 5–10% of previous torpor bout; *n* = 4) and late torpor (LT; Tb = 4°C for 80–95% of prior torpor bout, *n* = 4), early arousal (EAr; Tb = 5–12°C, *n* = 5), interbout-arousal (IBA; Tb = 37°C for 2–3 hr after Tb stabilization, *n* = 4), and entrance into torpor (Ent; Tb = 23–27°C, *n* = 5). Three groups consisted of animals in the non-hibernating portion of the year: summer active (SA; *n* = 5), collected in late July or early August, and two spring groups: (1) spring cold (SpC; *n* = 5) animals had spontaneously aroused from hibernation terminally, exhibiting no torpor for 10–20 days despite remaining in constant darkness at 4°C; and (2) a spring warm group (SpW; *n* = 6), at least 7 days after ambient temperature was raised first to 9°C for 5 days and then to 14°C. Animals were euthanized by exsanguination under isoflurane anesthesia, perfused with ice cold isotonic saline, and decapitated before dissection. Upon collection, BAT tissue was immediately snap-frozen in liquid nitrogen and then stored at −80°C.

### EDGE-tags

Total RNA was extracted from each BAT sample using the RNeasy Lipid Mini extraction kit (Qiagen, Venlo, the Netherlands) and assessed for quantity via NanoDrop and quality via the Bioanalyzer (RIN ≥ 8; Agilent Technologies, Santa Clara, CA). EDGE-tag libraries were created according to the protocol described by *Hong et al. (2011)* and submitted for massively parallel high-throughput sequencing at the genomic services lab at the HudsonAlpha Institute of Biotechnology, Huntsville, AL. The EDGE-tag libraries for all groups except ET and SA were sequenced on individual lanes of an Illumina GAIIx (Illumina, San Diego, CA). The ET and SA sample libraries were prepared subsequently;

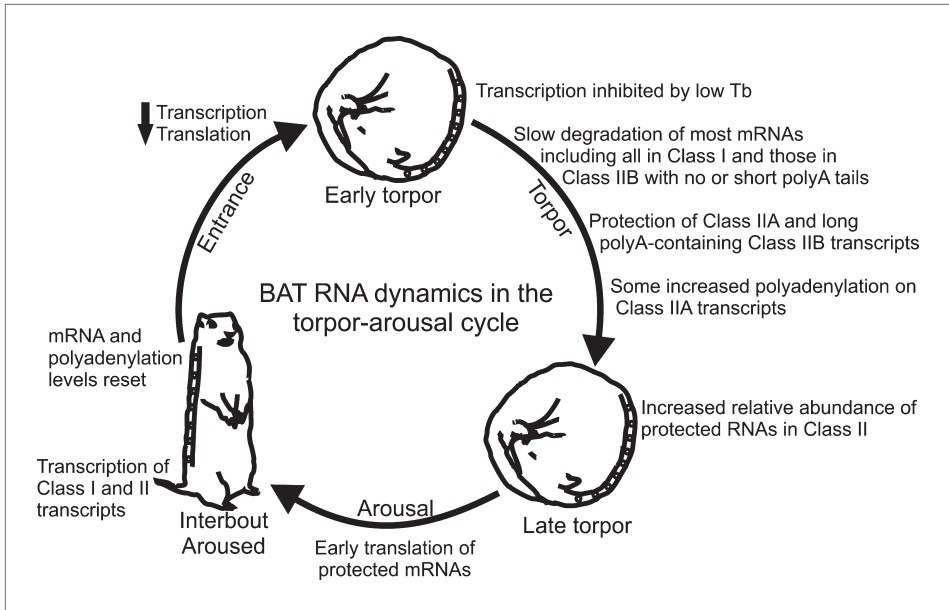

**Figure 7**. Model of BAT RNA dynamics in hibernation. Physiological stages of the torpor–arousal cycle are listed inside of the arrows and underneath representative animals. Key RNA changes are noted. See text for detailed explanation.

these were barcoded and all 10 samples sequenced on a single lane of an Illumina HiSeq, with two technical replicates added from the first round of sequencing.

## EDGE-tag read processing and annotation

The resulting reads were filtered for the presence of the *NlaIII* "CATG" site, trimmed of adapter sequences, and first aligned to the ground squirrel mitochondrial genome (*Hampton et al., 2011*) using Bowtie, allowing for two mismatches. Remaining unaligned reads were next aligned to the ground squirrel nuclear genome (Ensembl fasta v.71) using Bowtie with the same parameters. Resulting uniquely aligned nuclear reads were combined into read counts by tag position in the genome (requiring a minimum of 1 bp overlap). Poorly expressed tags were removed by converting the read counts into tags per million (TPM) counts for each library and requiring that at each tag position there will be a minimum of two TPM in *n*-1 samples in at least one sample group. The tags were annotated to their nearest known gene using the Ensembl annotations (.gtf, v. 71) or by homology to eutherian mammals in Ensembl; we required tags to be overlapping or within 3-kb downstream of the gene annotation to be considered annotated. Because we suspected that multiple, unannotated transcript isoforms of the same gene likely exist, tags that mapped to different positions within the same gene were not combined.

## Data analysis

Tags were normalized among all libraries using the full quantile method from the EDASeq (*Risso et al., 2011*) package in Bioconductor. There was a noticeable bias between the libraries sequenced on the GAIIx and the libraries later sequenced on the HiSeq; several steps were taken to remove tags that did not normalize well and contributed to this bias: (1) tags that differed by means of threefold between the technical replicates were removed; (2) remaining tags were then normalized by the full quantile method and tested for significant changes between sequencing platforms using the negative binomial GLM test in DESeq (*Anders and Huber, 2010*). Those with significant differences after a Benjamini–Hochberg false discovery rate correction (q < 0.05) were removed from the dataset. Finally, the remaining tags were again normalized using the full quantile method. The technical replicates were removed from all downstream analyses.

## Random Forests

The Random Forests using Variable Selection package (*Diaz-Uriarte, 2007*) first defined the subset of tags that produced the least amount of out-of-bag error in sample clustering. Here, 120,000 trees

were created in the initial forest, 10,000 trees created in the remaining iterations, and 20% of the variables dropped at each iteration. The selected tags then classified individual samples into groups using RF supervised clustering with 60,000 trees in R (*Svetnik et al., 2003*).

## Differential expression

Significant changes in tag expression among groups were detected by the negative binomial GLM test in DESeq. Both the test and null model included sequencing platform to discount changes introduced by library construction. The resulting p-values were adjusted for multiple testing with a Benjamini–Hochberg false discovery rate correction. Tags with a q-value of <0.05 were considered significant for differential expression among groups.

## Comparison with a previous BAT transcriptome dataset

A Pearson's correlation test was used to compare our mean tag expression values to the recent RNA-seq data of *Hampton et al. (2013)*, who identified 2,083/14,573 distinct transcripts differentially expressed among October, torpid, interbout-aroused and April 13-lined ground squirrels. Our late torpor, interbout-aroused and spring warm groups are roughly comparable to the last three of these, and 523 unique transcripts were identified as significant for differential expression in both datasets; these were tested for positive correlation ($r \geq 0.3$). Additionally, 1,466/2,083 significant transcripts detected by *Hampton et al. (2013)* were represented by tags in our study, of which 80% (1,170 transcripts) showed changes in the same direction (for at least one tag mapping to the same transcript; $r \geq 0.3$).

## DIANA clustering of tag expression patterns

The mean abundance value for each physiological group was first calculated for every significant differentially-expressed tag; these values were then mean-scaled. Pearson's correlation coefficients were calculated between every tag pair and were used to build a DIANA tree. The resulting tree was examined and cut at a height with the longest branches to produce individual DIANA clusters.

## Biological functional annotation

We assigned biological function to the tags within each DIANA cluster using the DAVID functional annotation clustering algorithm (*Huang da et al., 2009*). The functional annotation clusters were set at a medium or high stringency depending on which yielded the most biologically informative results. For each annotation cluster, the term with the most informative biological meaning was used. We also set the cut-off enrichment score of 1.3 for the term to be considered significantly enriched.

## RT-PCR

### cDNA synthesis

We first treated total BAT RNA from $n = 3$ samples in the IBA, LT, EAr, and SpW groups with DNase I (Invitrogen, Carlsbad, CA). A portion of this RNA was saved for direct conversion into cDNA while the rest was fractionated based on poly(A) tail length into short (≈25 nt) and long (>25 nt) poly(A) tail fractions using the PolyATtract mRNA Isolation System IV (Promega, Fitchburg, WI) as described (*Meijer et al., 2007*). The resulting RNA fractions were concentrated using the RNA Clean and Concentration Kit (Zymo Research, Irvine, CA). Random hexamer primed cDNA was synthesized using SuperScript III (Invitrogen). 3′ RACE cDNA was synthesized (*Peddigari et al., 2013*) with the reverse primer 5′-GCGAGCACAGAATTAATACGACTCACTATAGGTTTTTTTTTTTTTVN-3′. ePAT and TVN-PAT (TVN) cDNA (*Janicke et al., 2012*) from one poly(A) short and one poly(A) long RNA sample in each group was synthesized using the ePAT primer 5′-GCGAGCTCCGCGGCCGCGTTTTTTTTTTTT-3′; the TVN-PAT primer was the same, except that it had a (VN) added to the 3′ end.

### PCR

Gene-specific primers were designed and purchased using the ground squirrel transcript sequences in Ensembl and the PrimerQuest design tool (IDT, Coralville, IA; www.idtdna.com). Where possible, the primer pairs encompassed an EDGE-tag of interest (*Supplementary file 1A*); these were chosen using several criteria, including fold change, q-value significance, correlation with *RPPH1*'s expression pattern, and biological relevance to BAT function. Random hexamer and 3′ RACE primed cDNA were amplified using FastStart Taq (Roche, Basel, Switzerland) with either the specific primer set or the 3′ RACE universal reverse primer (3′ RACE cDNA; 5′-GCGAGCACAGAATTAATACGACT-3′). The resulting

amplicons were separated in 2% agarose 1× TAE gels and visualized with Sybr-safe DNA gel stain (Invitrogen); the gel bands were imaged with a Typhoon scanner (GE Healthcare, Pittsburg, PA) and analyzed using ImageQuant TL software (GE Healthcare).

In order to confirm fractionation of RNA based upon poly(A) tail length, one TVN and all ePAT cDNA samples were amplified with the creatine kinase brain (*CKB*) forward primer (*Supplementary file 1A*) and the ePAT universal reverse primer 5′-AGCTCCGCGGCCGCG-3′; the resulting amplicons were visualized and analyzed as described above. The ePAT band sizes were subtracted from the TVN band size to estimate the poly(A) tail length of each sample (*Figure 4—figure supplement 1*).

## PCR amplicon cloning and sequencing
The PCR amplified bands were cut from gels and DNA purified using a mini-elute gel extraction kit (Qiagen). Between 1–4 µl of eluted PCR product were inserted into pCR4-TOPO-TA or pCR2.1-TOPO-TA (Invitrogen) and transformed into TOP10 Chemically Competent *Escherichia coli* cells (Invitrogen). Plasmids containing an insert of the correct size were sequenced using M13 forward or reverse primers. Sequences were aligned to the *Ictidomys tridecemlineatus* genome using BLASTN or BLAT in Ensembl.

## Quantitative PCR
RT-qPCR was performed using a StepOnePlus instrument (Applied Biosystems, Foster City, CA) with FastStart Universal SYBR Green Master (ROX; Roche). All biological samples were measured in triplicate using a standard curve specific for each transcript/primer set. Outlier measurements were removed and transcript abundance values were calculated for all biological samples using the mean of the technical replicates for each sample. To correct for sample loading inaccuracies using a method independent of normalization to a housekeeping gene (which assumes constant expression of that gene), we instead created a custom normalization factor for each biological sample based upon the sample's overall tendency to be higher or lower for transcript expression within its own respective sample group (e.g., within EAr or within LT). We first calculated each transcript's fold change for a given sample relative to the median transcript abundance measurement within its respective sample group. We then created a normalization factor for each biological sample by calculating the mean of that sample's within-group transcript fold change across all transcripts. Each of the individual transcript abundance measurements were next divided by their respective sample's normalization factor. After normalization, transcripts were tested for significant expression changes among sample groups via one-way ANOVA (α = 0.05) and for correlation with their corresponding EDGE-tag using a Spearman's rank correlation test.

## Western blot
Protein homogenates were prepared from BAT axillary tissue as described (*Hindle and Martin, 2014*) from *n* = 3 samples for each IBA, LT, EAr, and SpW groups. Western blots were used to measure PNPLA2 (1:1000, rabbit pAb #2138, Cell Signaling Technology, Danvers, MA) and β-tubulin (1:1000, goat pAb #ab21-57, Abcam, Cambridge, MA), both detected using IRdye-conjugated secondary antibodies (1:20,000 IRDye 800CW anti-rabbit and 1:10,000 IRDye 680LT anti-goat; Li-Cor, Lincoln, NE). Proteins were imaged (Odyssey near-infrared imaging system, Li-Cor) and analyzed with ImageQuant TL software (GE Healthcare). To correct for inconsistencies in protein loading, each PNPLA2 band was normalized to the β-tubulin band within the same lane. Changes in PNPLA2 abundance among the sample groups were assessed by a one-way ANOVA (α = 0.05).

## Mathematical modeling of transcript dynamics
To investigate RNA transcript dynamics across a torpor–arousal cycle, we developed a differential equations-based mathematical model simulating the abundance of 1,400 bulk and 50 protected transcripts. This ratio of bulk Class I transcripts to protected Class II transcripts reflects the ratio observed in the data (14,267 bulk; 531 protected in DIANA Clusters 5 and 6, *Figure 2D*). All modeling and model analysis were performed in MATLAB (MathWorks, Natick, MA). The mathematical model is described in detail in *Supplementary file 2*.

## Motif discovery and enrichment
We examined the transcripts that appeared to increase in torpor for shared motif enrichment in their 3′ UTRs. Due to uncertainty in 3′ UTR structure and length, we included only transcripts that contained significantly differentially expressed (D.E.) tags <1 kb 3′ to their nearest protein coding feature. First,

the transcripts were divided by whether they fell into the 'torpor-increased' DIANA clusters 4–6 or 'torpor-decreased' DIANA clusters 1–3. Each transcript was counted only once; in cases where multiple tags mapped to the same transcript, the most 3' D.E. tag was used for assignment of the transcript to a particular cluster. Annotation of the 3' UTRs in the ground-squirrel genome is currently sparse and likely imprecise; thus, for each gene, we conservatively defined the '3' UTR' to be the 500-nt region immediately 3' to the stop codon. To identify enriched motifs, we used the motif discovery algorithm EXTREME (*Quang and Xie, 2014*) with the default settings, except allowing 0 gaps, and with the 500-nt '3' UTR' sequences from 'torpor-increased' transcripts input as the positive set and the 500-nt '3' UTR' sequences from the 'torpor-decreased' transcripts input as the negative set. We considered resulting motifs with E values <1 as significantly enriched in the positive set. The enriched motifs were used as input in the MAST tool (*Bailey and Gribskov, 1998*) of MEME Suite (*Bailey et al., 2009*), which counted the number of transcripts containing the motif (E-value <10) within each DIANA cluster. As a control, we also counted the number of motif occurrences in the 500-nt '3'UTRs' of 484 non-significant D.E. transcripts (q > 0.97; 500 transcripts were originally chosen but 16 lacked 3' sequence data, hence 484 transcripts). Finally, to detect motifs enriched in all clusters, 3' UTR sequences from 20 randomly chosen transcripts in each cluster (120 total) were compared against a scrambled background set in MEME (*Bailey and Elkan, 1994*), with the settings set at a maximum width of 8-nt and a search of the given strand only.

To identify the putative RNA binding proteins that might recognize these motifs, the significantly enriched motifs were uploaded into the TOMTOM motif comparison tool (*Gupta et al., 2007*) of the MEME Suite. The database against which these motifs were first searched consisted of the RNA binding protein motifs described by *Ray et al. (2013)*; however, no significant matches were found. We next added the nine C-rich motifs reported and provided by *Ji et al. (2013)* to the RNA binding protein motif database and repeated the search for significantly enriched motifs. The significance values for motif matches were calculated via Pearson correlation coefficient in TOMTOM. All motif logos were generated in TOMTOM.

## Acknowledgements

We thank S Clark, C Henegar, A Hindle, J Spalding, and A Zukowski for help with experiments and useful discussion. We thank J Wan, X Ji, Y Xing, and SA Liebhaber for their assistance in providing the PCBP motif files.

## Additional information

### Funding

| Funder | Grant reference number | Author |
| --- | --- | --- |
| National Institute of Diabetes and Digestive and Kidney Diseases | R21DK095180 | Sandra L Martin |
| Division of Mathematical Sciences | 1121361 | Cecilia Diniz Behn |

The funders had no role in study design, data collection and interpretation, or the decision to submit the work for publication.

### Author contributions

KRG, conceived the project, generated and analyzed EDGE, RT-qPCR and western blot and motif data and drafted the manuscript, Conception and design, Acquisition of data, Analysis and interpretation of data, Drafting or revising the article; CDB, developed and implemented the mathematical modeling, Acquisition of data, Analysis and interpretation of data, Drafting or revising the article; GSB, assisted with EDGE-tag study design, EDGE data generation, analysis and interpretation, and assisted with revising of manuscript, Acquisition of data, Analysis and interpretation of data, Drafting or revising the article, Contributed unpublished essential data or reagents; JRH, assisted with processing and annotation of EDGE-tag libraries, interpretation of data and writing of manuscript, Acquisition of data, Analysis and interpretation of data, Drafting or revising the article; SLM, directed

the project, interpreted data and drafted manuscript with KRG, Conception and design, Analysis and interpretation of data, Drafting or revising the article, Contributed unpublished essential data or reagents.

### Ethics

Animal experimentation: This study was performed in strict accordance with the recommendations in the Guide for the Care and Use of Laboratory Animals of the National Institutes of Health. All animals were handled according to the Institutional Animal Care and Use Committee protocol 44309(03)1E approved by the University of Colorado. All surgeries and euthanasia were performed under isoflurane anesthesia, and every effort was made to minimize suffering.

## Additional files

### Supplementary files

• Supplementary file 1. RT-qPCR Measurements. (A) Oligonucleotide primers used in RT-qPCR. (B) Spearman's rank correlations of RT-qPCR abundance measurements to EDGE-tag abundance measurements. (C) One-way ANOVA p-values of RT-qPCR measured RNA abundance changes among sample states.

• Supplementary file 2. Mathematical Modeling of Transcript Dynamics.

### Major dataset

The following dataset was generated:

| Author(s) | Year | Dataset title | Dataset ID and/or URL | Database, license, and accessibility information |
| --- | --- | --- | --- | --- |
| Grabek KR, Diniz Behn C, Barsh GS, Hesselberth JR, Martin SL | 2014 | Data from: Enhanced stability and polyadenylation of select mRNAs support rapid thermogenesis in the brown fat of a hibernator | doi: 10.5061/dryad.5hh54 | Available at Dryad Digital Repository under a CC0 Public Domain Dedication. Contains: Supplementary file 3. (A) EDGE-tag raw counts. (B) EDGE-tag normalized counts. (C) EDGE-tags significant for changed expression. (D) List of All DAVID Functional Annotation Clusters |

**Reporting standards:** Standard used to collect data: The dataset published is in accordance with the guidelines listed in the required minimum information about a high-throughput nucleotide sequencing experiment, MinSeqe.

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
