## [Decision Letter]

Thank you for sending your work entitled “Enhanced stability and polyadenylation
of select mRNAs support rapid thermogenesis in the brown fat of a hibernator” for
consideration at *eLife*. Your article has been evaluated by Randy
Schekman (Senior editor) and, a Reviewing editor, and 2 reviewers.

The Reviewing editor and the reviewers discussed their comments before we reached this
decision, and the Reviewing editor has assembled the following comments to help you
prepare a revised submission.

This manuscript investigates the transcriptome dynamics of hibernating ground squirrels.
The authors performed RNA-Seq profiling of brown adipose tissue (BAT) in ground
squirrels across different periods of the torpor-arousal cycle. Three major observations
were made: (1) unsupervised clustering of the samples revealed three main
groups—spring, winter warm, and winter cold. Between these three groups, 1638
genes with significant differential steady-state transcript levels were detected. Gene
ontology analysis identified processes associated with BAT function in the winter
groups; (2) a subset of transcripts in the winter groups reached their highest abundance
during early torpor, late torpor, and early arousal despite suppressed transcription
rates at low body temperatures. RT-PCR analysis revealed two classes of transcripts
during torpor: transcripts were either degraded and not replenished, or they were
stabilized by the maintenance or acquisition of a long poly(A) tail; (3) the authors
propose a mathematical model to describe the changes in transcription rates and
degradation could affect differential gene expression in torpor. A temperature-dependent
mechanism was identified that protected a subset of transcripts relative to bulk RNA
degradation.

This study provides novel insight into the transcriptional dynamics of hibernating
animals. In particular, how certain transcripts are activated/maintained despite
suppressed transcription during low body temperatures is poorly understood. The findings
in this study would appeal to those interested in hibernation dynamics, as well as those
who study post-transcriptional regulation. While it is known that polyadenylation is
important for mRNA stability, localization, and translation, the authors present
evidence for the first time that this mechanism is implicated in the torpor-arousal
cycle.

Overall, the manuscript reads very well written with clearly reported observations and
results. A significant limitation of the manuscript, as was discussed by the three
reviewers, is the lack of experimental data addressing the molecular mechanisms
underlying the authors' observations.

Main comments:

1) The authors' manuscript would be significantly strengthened if they could
provide mechanistic insight into the temperature-dependence of poly(A) tail length. For
example, the reviewers wonder whether BAT samples from torpor-arousal cycling animals
could be used to assess differential cross-linking/binding of PABP1 and/or RBM3 to
transcripts that display polyA length changes, to test the authors' proposal (based
on previous observations they have cited) that these factors may be associated with
differential transcript stability dynamics. This experiment could potentially be
performed using CLIP-Seq, or CLIP-qRT-PCR on selected transcripts, by employing
commercial antibodies that bind to conserved epitopes on these proteins (which overall
are highly conserved). Related to this, have the authors examined the levels of
transcripts from genes that encode factors involved in the control of polyA tail length
(i.e. deadenylation or polyadenylation factors) across their RNA-Seq datasets? Such
information may lead to mechanistic insight as well.

2) The authors are requested to provide basic information on polyA site usage in the
different samples. How many sites were detected and how does the rate of detection of
polyA site usage relate to the steady-state expression of the corresponding
transcripts?

3) The authors should provide a more basic description (i.e. for the general reader) of
how the mathematical model they developed was conceived and how well it reliably models
the experimental data. In this regard, it is also not clear in the main text or Methods
how levels of bulk and protected transcripts were determined.

---

## [Author Response]

*1) The authors' manuscript would be significantly strengthened if they
could provide mechanistic insight into the temperature-dependence of poly(A) tail
length. For example, the reviewers wonder whether BAT samples from torpor-arousal
cycling animals could be used to assess differential cross-linking/binding of PABP1
and/or RBM3 to transcripts that display polyA length changes, to test the
authors' proposal (based on previous observations they have cited) that these
factors may be associated with differential transcript stability dynamics. This
experiment could potentially be performed using CLIP-Seq, or CLIP-qRT-PCR on selected
transcripts, by employing commercial antibodies that bind to conserved epitopes on
these proteins (which overall are highly conserved). Related to this, have the
authors examined the levels of transcripts from genes that encode factors involved in
the control of polyA tail length (i.e. deadenylation or polyadenylation factors)
across their RNA-Seq datasets? Such information may lead to mechanistic insight as
well*.

We agree that our manuscript could be strengthened by providing mechanistic insight into
the temperature-dependence of poly(A) tail length and selective transcript stability.
Although the suggested strategy of either CLIP or CLIP-Seq using antibodies to PABP1 or
RBM3 might enhance our understanding of the molecular mechanism involved, this is often
not a trivial undertaking in ground squirrels in terms of identifying useful antibodies.
We do not think that evidence for PABP1 binding would be particularly helpful because it
is unlikely to provide the specificity needed to explain the selectivity among
transcripts that is evident in our data. Before embarking on a potentially difficult
search for an appropriate immunoprecipitating RBM3 antibody, we employed a
motif-searching bioinformatics approach to see whether RBM3 binding sites were enriched
in the protected transcript subset. Although neither of the two (worrisomely unrelated)
RBM3 binding sites that have been reported by others are significantly enriched in our
population of protected transcripts (see Figure 8), we believe that the motif searching strategy employed set us on a
path that has proven to be extremely fruitful. Specifically, we detected two C-rich
motifs that are significantly enriched in the 3’ regions of the stabilized
transcripts; these motifs were the only enriched motifs (see figure) and are predicted
to be bound by a poly(C) binding protein (PCBP). Significantly, the mRNA expression for
one of the PCBP family members, PCBP3, increased during the winter in our dataset, in
contrast to most of the other RNA binding proteins that were differentially expressed.
We document this motif and its link with PCBP3 in a new Figure in the manuscript (Figure 6) and now suggest, based on the motif
enrichment analysis, that a PCBP paralog, most likely PCBP3, may be mechanistically
linked to the observed transcript stability. We believe this analysis adds significant
mechanistic insight underlying mRNA stabilization and polyadenylation and paves the way
for the next step, which would be to identify the RNAs that are bound by PCBP3 (or its
paralogs) at different hibernation timepoints using CLIP-Seq. We would like to argue
here, however, that such a next step is well beyond the scope of this study.Author response image 1.Motif comparisons: Left side, RBM3 motifs reported by (top) Liu, Y. et al.
(2013) Sci. Rep. 3, doi:10.1038; (bottom); Ray, D. et al. (2013), Nature 499,
172. Right side, top four motifs in the hibernation-increased subset of RNAs in
our dataset. Note only the top two, as reported in the new Figure 6 of our manuscript, are significantly
enriched.

2) The authors are requested to provide basic information on polyA site usage in
the different samples. How many sites were detected and how does the rate of
detection of polyA site usage relate to the steady-state expression of the
corresponding transcripts?

We hesitate to make any inference regarding poly(A) site usage at this point. If
*NlaIII* cut to completion, we at best have identified the 3’
most “CATG” site within each transcript, not the 3’ end itself.
Thus, we do not think these tags should be viewed as a conclusive proxy for defining
3’UTR’s and poly(A) site usage. However, we do provide all tag locations
within their respective transcripts in the supplementary tables, so that others doing
poly(A)-Seq analysis on ground-squirrel transcripts can easily make such comparisons.
Our observations of enhanced mRNA stabilization and polyadenylation during torpor were
an unanticipated finding from the digital transcriptome data that were collected to
identify differentially-expressed genes linked to the phenotypic dynamics of
hibernation. Although in the process we have detected what we suspect are alternative
polyadenylation sites, focusing on them here would detract from the main findings of
this work.

*3) The authors should provide a more basic description (i.e. for the general
reader) of how the mathematical model they developed was conceived and how well it
reliably models the experimental data. In this regard, it is also not clear in the
main text or Methods how levels of bulk and protected transcripts were
determined*.

We made several changes throughout the manuscript to enhance this information in the
main text.